# Heart Sound Classification Based on Multi-Scale Feature Fusion and Channel Attention Module

**DOI:** 10.3390/bioengineering12030290

**Published:** 2025-03-14

**Authors:** Mingzhe Li, Zhaoming He, Hao Wang

**Affiliations:** 1Research Center of Fluid Machinery Engineering and Technology, Jiangsu University, Zhenjiang 212013, China; 2212211038@stmail.ujs.edu.cn; 2Department of Mechanical Engineering, Texas Tech University, Lubbock, TX 79411, USA; zhaoming.he@ttu.edu; 3School of Electrical and Information Engineering, Jiangsu University, Zhenjiang 212013, China

**Keywords:** heart sound classification, channel attention, feature fusion, transfer learning

## Abstract

Intelligent heart sound diagnosis based on Convolutional Neural Networks (CNN) has been attracting increasing attention due to its accuracy and efficiency, which have been improved by recent studies. However, the performance of CNN models, heavily influenced by their parameters and structures, still has room for improvement. In this paper, we propose a heart sound classification model named CAFusionNet, which fuses features from different layers with varying resolution ratios and receptive field sizes. Key features related to heart valve diseases are weighted by a channel attention block at each layer. To address the issue of limited dataset size, we apply a homogeneous transfer learning approach. CAFusionNet outperforms existing models on a dataset comprising public data combined with our proprietary dataset, achieving an accuracy of 0.9323. Compared to traditional deep learning methods, the transfer learning algorithm achieves an accuracy of 0.9665 in the triple classification task. Output data and visualized heat maps highlight the significance of feature fusion from different layers. The proposed methods significantly enhanced the performance of heart sound classification and demonstrated the importance of feature fusion, as interpreted through visualized heat maps.

## 1. Introduction

### 1.1. Background

Heart sound classification, which focuses on distinguishing heart sounds collected from patients with cardiovascular diseases (CVDs) and analyzing their specific types, is one of the most effective and non-invasive methods for the early detection and diagnosis of CVDs.

CVDs are a group of diseases that include heart valve disease, heart failure, and hypertension, which contribute significantly to premature death rate worldwide. According to the World Health Organization, approximately 17.9 million people died from various kinds of CVDs in 2019, representing 32% of all global deaths [1]. In China, it is estimated that a total of 5.09 million people died from CVDs in 2019, with an age-standardized mortality rate of 2.76‰ [2]. Due to the acceleration of social pace and changes in lifestyle, the morbidity of CVDs is still increasing. The causes of CVDs are complex and variable, the onset is rapid, and the mortality rate is extremely high. Therefore, developing heart sound auscultation methods for rapidly and accurately diagnosing CVDs has become increasingly important.

Heart sound originates from vibration while heart valves and vascular wall struck by blood in systole and diastole reflects hemodynamic conditions of the heart. Therefore, heart sound is a significant metric for diagnosing CVDs. A cycle of heart sound signal can be divided into four components [3], which are shown in Table 1. Fundamental heart sound is crucial for diagnosing CVDs. Generally, extra low-frequency murmurs occur with fundamental heart sound and are a sign of heart valve diseases, which require further diagnosis.

At present, traditional heart sound diagnosis relies on the experience of clinical experts. Previous studies have shown that the precision of traditional diagnosis is about 81% to 83% [4]. Therefore, developing methods for computer-aided heart sound diagnosis is significant for clinics.

Deep learning (DL) is used as a tool to solve the above problem. As a main branch of machine learning, DL aims at solving the difficulty of modeling by constructing complex theories using simple theories. Nowadays, methods of deep learning, especially CNN, are widely used for image recognition and audio processing. In the subdivision of heart sound diagnosis, DL attracts increasing attention. Most prior works have focused on adapting image processing models for time series classification, with minor modifications or the construction of new architectures to continuously optimize classification performance, which are aimed at enhancing accuracy and robustness while reducing parameters and computational costs.

### 1.2. Relative Works

Nowadays, most studies that focused on heart sound classification effectively enhanced the performance of models. This part focuses on works related to feature fusion, attention mechanism, and transfer learning.

In recent years, multi-scale feature fusion of CNN models was widely researched and applied for image processing. For example, Feature Pyramid Networks (FPNs) [5] proposed by Liu et al. and the skip-connection model [6] proposed by Shrivastava et al. have been widely used. Present studies show that fusing feature maps from different layers with different resolution ratios into one layer can integrate features from deeper layers and upper layers, improving accuracy and generalization ability significantly.

Tschannen et al. [7], the early users of the feature fusion method, presented a novel classification method fused features extracted by CNN, state statistics, and Power Spectral Density (PSD) in 2016. Li et al. [8] fused time domain features extracted by a Gated Recurrent Unit (GRU) and frequency-domain features extracted by a group convolutional layer and expanded the feature space.

U-Net, based on skip-connection, was applied to heart sound detection [9,10]. Taresh et al., the first to apply the feature fusion mechanism to heart sound auscultation, proposed a U-Net-based model that decodes features from deep layers by upsampling and fuses it with top layers for heart sound denoising. Xu et al. [11] constructed HMSNet, the first one introduced for the theory of multi-scale feature fusion into heart sound classification, which has the advantages of different hierarchical architecture for improving the performance of algorithms. The local test results showed that HMSNet performed approximately 1% better than traditional ResNet. PCTMF-Net [12], proposed by Wang et al., was built using a two-way parallel CNN module that can fuse feature maps processed by a CNN and Transformer, and better learn information from different convolutional layers, which increases the accuracy of heart sound classification result to 93%.

Channel attention mechanism (CAM), simulating human vision paying more attention to key signs, was successfully applied to the field of computer vision. By focusing on the most informative regions within feature maps, CAM enables models to capture key features of signals more precisely. Tian et al. [13] firstly introduced CAM to heart sound classification and proposed a model adding Efficient Channel Attention (ECA) block. More and more research concerning the application of CAM in heart sound detection [14,15,16,17] has shown that CAM can significantly improve the accuracy and robustness of models.

Currently, most studies focus on binary classification due to the lack of heart sound datasets labeled with different types of CVDs. Yaseen et al. [18] created a dataset labeled with five classes, whose signals are high-SNR. The dataset is widely used for the research of heart sound multi-classification. However, classification algorithms are still limited by the problems of datasets. The main issues with heart sound datasets include sample imbalance and insufficient dataset size for deep learning training. Due to the large number of parameters in deep learning models, the lack of sufficient training data can negatively impact the model’s generalization ability, thereby affecting the final training performance. In order to solve problems caused by datasets, non-traditional machine learning, including transfer learning, unsupervised learning, and semi-unsupervised learning, was introduced to improve research on heart sound datasets. Most studies on transfer learning focus on heterogeneous transfer learning, which depends on large-scale pretraining datasets and weighted models to break limitations of heart sound datasets. For example, both Koike et al. [19] and Malty et al. [20] applied AudioSet, a large-scale audio dataset, for the source domain to propose a transfer learning method of heart sound classification, which transferred weights between time series. Mukherjee et al. [21] proposed a transfer learning method using the ImageNet dataset as the source domain to diagnose heart valve diseases based on heart sound spectral features.

At present, most studies on heart sound classification algorithms employ end-to-end feature extraction methods based on raw heart sound signals. The performance of end-to-end methods depends heavily on the structure and parameters of the models. The robustness of deep learning models to noise and imbalanced datasets still has room for improvement, requiring further advancements in generalization, which is significant to improve the feasibility in clinical practice.

Furthermore, the lack of larger datasets, especially datasets with more diverse disease types, has hindered the validation of previous algorithms on a broader range of cardiovascular diseases and populations. Non-traditional machine learning methods can partially address these issues. However, research on heart sound classification based on non-traditional machine learning methods is still relatively scarce, particularly in the area of homogeneous transfer learning methods.

### 1.3. Proposed Works

In this paper, we combined CAM with multi-scale feature fusion with the proposed CAFusionNet. We fetched feature maps from three pooling layers and applied Gated Channel Transformation (GCT) as attention blocks to weight each feature map. All feature maps weighted by GCT blocks were finally fused in the deepest layer, which aims at combining pathological features from different resolution ratios and improve the performance of classification algorithms. We applied heat maps based on gradient information to visualize and compare them with feature maps at different depths, showing importance of feature fusion. In order to solve problems about datasets labeled with different types of CVDs, we tried to propose homogeneous transfer learning to carry weights trained by heart sound binary classification tasks. A local test, held with public heartbeat sound datasets and datasets collected by our group, shows that CAFusionNet achieves an accuracy of 0.9323 in binary classification, performing better than present models and traditional residual model. The accuracy of the transfer learning methods reaches 0.9665, which is better than the traditional deep learning method.

The proposed model enhances the clinical applicability of intelligent heart sound diagnosis systems. Improvements in performance metrics and the use of visualized heat maps make intelligent heart sound diagnosis both more feasible and interpretable. Transfer learning has significance in the intelligent diagnosis of specific CVDs in clinical practice.

The contents of this paper are organized as follows: The proposed methods including the CNN-based CAFusionNet and transfer learning methods are proposed in Section 2. The process, methods, and results are shown in Section 3. In Section 4, we expose a discussion of our present research. In Section 5, we make a conclusion of this paper and discuss future expectations.

## 2. Materials and Methods

### 2.1. Datasets and Environment

Any training tasks based on deep learning rely on large-scale datasets. To some degree, the scale and quality of datasets impact the generalization ability of the training models. In this paper, we used two datasets that are shown in Table 2 in our research. The first dataset (dataset A) is PhysioNet/CinC Challenge 2016 [22], which includes 3240 heart sound records within binary classes (normal/abnormal) from 9 sub-datasets. All sub-datasets were collected from different hospitals, instruments, and environments, which enriches the complexity of signals and provides a comprehensive feature set for research. The second dataset (dataset B) was created by our group [23], which includes heart sound records of 86 pediatric patients from 4 months to 16 years, whose clinical diagnosis can be divided into normal, Atrial Septal Defect (ASD), and Ventricular Septal Defect (VSD). In the binary task, all samples labeled with ASD or VSD in dataset B were seen as abnormal signals.

The preprocessing algorithms were performed on a computer with Windows 10 (x64) powered by Intel(R) Core(TM) i5-11400. The preprocessing tasks of all the series were performed by Matlab 2021A. After preprocessing, all series were copied to another computer with Windows 11 (x64) powered by an NVIDIA RTX 2080Ti GPU and an Intel(R) i9-9920X CPU to finish the training process. The software environment for the training and test included Python 3.10, as well as TensorFlow 2.9.1 and Keras 2.9.0, for constructing deep learning models.

### 2.2. Preprocessing

Preprocessing of the heart sound signals includes resampling, normalization, denoising, and segmentation, which aims at ensuring the quality of the signals. At first, we resampled all signals to 1000 Hz and normalized all signals by z-score, which prevents the feature space from the impact of non-unified amplitudes and frequencies. Then, we used Butterworth band-pass filters for denoising to remove high-frequency noise while reserving low-frequency pathological murmurs. Following the experience of recent studies, we set the ceiling and floor of the passed frequency as 400 Hz to 25 Hz.

The number of cardiac circles are different between the samples in the PhysioNet/CinC 2016 dataset. Therefore, segmentation is required before training to normalize the feature space of all the samples. We used logistic regression proposed by Springer et al. [24] to segment the denoised signals and made every 4 cycles per samples to be the dataset for training.

### 2.3. Models

The model proposed in this paper is based on ResNet proposed by He et al. [25] in 2015. The fundamental unit of ResNet is shown in Figure 1, which connects adjacent convolutional layers directly. This mechanism resolved the problem of vanishing and exploding gradients while stacking more layers and simplified the computational complexity. We applied the Rectified Linear Unit (ReLU) as the activation function, which aims at enhancing nonlinear characters of the model. L2 normalization is applied to reduce the over-fitting problem.

This paper constructed the model by five residual blocks with max pooling layers for down sampling. To contribute to down sampling, the receptive field is expanded and the complexity of the algorithm is reduced. The latest layers are two full-connect layers and the activation functions are ReLU and sigmoid. The theory of a single residual block can be described as (1), while $F(x,w_{i})$ represents convolutional layers and activation functions in the residual block, and $w_{i}$ represents the trainable weights of each layer.(1)$$y=F(x,w_{i})+x$$

In residual networks, the size of the receptive field increases with the depth of the layers. The increment of the receptive field per convolutional layer is shown in (2).(2)$$RF_{i}=s_{i}\left(RF_{i-1}+\left(k-1\right)a_{i}\right)$$

While *k* is the kernel size and we set *k* = 3 in this paper; $a_{i}$ and $s_{i}$ are the dilation rate and stride size of the *i*-th convolutional layer. In addition, the receptive field can be expanded to n times by pooling layers (size = n). In general, the dilated convolutional layers and pooling layers effectively expand the receptive field size of the proposed model. The receptive fields of every layer are shown in Table 3.

As shown in Table 3, after passing through five residual blocks and their corresponding pooling layers, the receptive field size has increased significantly, providing the network with enough capacity to recognize and distinguish between different heart sound categories. The differences in receptive field sizes at various locations in the residual network allow us to understand the network’s ability to capture and abstract signal features at different depths. In residual networks, feature map outputs by the top convolutional layers are characterized by high spatial resolution but exhibit weak semantic representation due to the lack of substantial processing. In contrast, feature maps generated by deeper layers contain abstract and strong semantic representation capabilities but have low temporal resolution, with some features being lost. To address these limitations, merging the feature maps from different layers and merging feature maps from different network layers enables the model to leverage complementary strengths, compensates for inherent weaknesses, and enriches the overall feature information in the output. Specifically, the fusion mechanism in our model integrates feature maps extracted from the 3rd, 4th, and 5th residual blocks. The methods for fusing feature maps include addition and concatenation. Adding feature maps can integrate feature information without expanding the feature size, but it may lose some key details. In contrast, concatenating feature maps can effectively expand the feature space without losing any information. Therefore, we used a concatenation layer to fuse features from the three aforementioned layers.

This hierarchical approach shown in Figure 2 enhances the model’s ability to capture both fine-grained details and high-level abstractions, improving its robustness and representational capacity across varying tasks.

Inspired by naked eyes paying attention to key points of vision, the attention mechanism (AM) is proposed to increase the weights of key features. The Squeeze-and-Excitation Network (SENet), the pioneered channel attention model, is constructed by a squeeze block that collects global information and an excitation block which captures channel-wise relationships and outputs an attention vector [26]. Recent studies show that SENet can significantly improve the robustness of models and the precision of training results by weighting the features of different channels. However, the excitation block of SENet constructed by fully connected layers increases the complexity of the models, especially while training huge datasets. Otherwise, the proposed model needs numerals of channel attention blocks in this paper, thus, choosing a light-weight attention module is crucial. This paper uses GCT as the attention module to capture key features of the 3 layers mentioned above, where, in this way, the proposed model reduces the parameters significantly. Compared to SENet, GCT contains only 3 trainable parameters. The main theory of GCT is shown in (3) and (4).(3)$$Y=X+X\cdot \tanh\left(\frac{\alpha \cdot Norm\left(X\right)\cdot \sqrt{C}}{Norm\left(X\right)}\cdot \gamma +\beta \right)$$(4)$$Norm\left(X\right)=\frac{\sqrt{C}X}{{\left\|X\right\|}_{2}}=\frac{\sqrt{C}X}{\sqrt{\sum_{i=1}^{C}X_{I}+\varepsilon }}$$

While *C* represents time domain length of feature maps, *α*, *γ*, and *β* are trainable parameters that represent channel weights, gated weights, and bias. Both *γ* and *β* control the size of the gate, which compose the excitation module. tanh(·) represents the tahn activation function, which can reflect the output feature to [−1,1].(5)$$\tanh(X)=\frac{e^{x}-e^{-x}}{e^{x}+e^{-x}}$$

Compared to *sigmoid*, the output of tanh makes training metrics converge faster, while the feature map of the heart sound signal are center in 0.(6)$$sigmoid\left(x\right)=\frac{1}{1+e^{-x}}$$

The squeeze module consists of Local Response Normalization (LRN), which can conform global information of feature maps without losing extremum. The whole process of LRN is shown in (4), while $\varepsilon =10^{-5}$. The structure of a GCT block is shown in Figure 3.

### 2.4. Gradient-Weighted Class Activation Mapping (Grad-CAM)

Grad-CAM, based on gradient information of each layer in the model, is an important method for visualizing CNN models. It aims at improving the interpretability of deep learning models by marking significant points that make sense for the final predicted results in feature maps. The steps of calculating Grad-CAM after a certain layer can be described as follows:Choose the sample in the validation set with the lowest BCE value.

Calculate the BCE values of each sample the in validation set with the predicted labels and true labels. The sample with the lowest BCE value is the best sample, used for input feature with its true label.(7)$$Loss=-\frac{1}{N}\sum_{i=1}^{N}\left[y_{True}\cdot \log\;y_{Predict}+\left(1-y_{True}\right)\cdot \log\left(1-y_{Predict}\right)\right]$$

While $y_{True}$ is one-hot encoding of true label and $y_{Predict}$ is the probabilistic interpretation of the predicted results.

2.Calculate the gradient value by the selected sample.

Input the best sample sequence into the model after 150 training iterations for forward propagation. Through the CNN network and task-specific computation, the raw score $y_{c}$ of the disease diagnosis result is returned. At this point, the initial gradient for the target class (normal/abnormal) of this classification task is set to 1. Through backward propagation of $y_{c}$, calculate the gradients in the certain layer by the following functions.(8)$$\alpha_{k}^{c}=\frac{1}{N}\sum_{i=1}^{N}\frac{\partial y^{c}}{\partial A_{i}^{k}}$$

While $A_{i}^{k}$ is the *i*-th value of the k-th channel in visualized the layer (Target Layer), $\frac{\partial y^{c}}{\partial A_{i}^{k}}$ represents the gradient information through backward propagation, and $\alpha_{k}^{c}$, the weight of the certain class (normal/abnormal) in the *k*-th channel, is returned.

3.Combine the gradient values with the channels with weights.


(9)
$$Grad_{c}=ReLU\left(\sum_{k}\alpha_{k}^{c}A^{k}\right)$$


$A^{k}$ represents the forward propagated feature map in the *k*-th channel of the target layer.

### 2.5. Multi-Classification Based on Transfer Learning

At present, there are limited and insufficiently scaled heart sound datasets labeled with disease classifications, which rely on precise annotations provided by clinical experts, making the data difficult to obtain and challenging to ensure consistent quality. Additionally, on small-scale datasets, the limited number of training samples makes it challenging for models to learn sufficient feature representations, thereby affecting their generalization capabilities and performance. Therefore, transfer learning is proposed to break the bottlenecks that originated from the scale of the datasets. According to the distance of feature space between the source domain and target domain, transfer learning can be classified as homogeneous transfer learning and heterogeneous transfer learning. Homogeneous transfer learning refers to scenarios where the source domain and the target domain share the same or highly similar structures and distributions in the feature space. In such cases, the model parameters and knowledge learned from the source domain can be directly transferred to the target domain since the data from both domains exhibit similar feature representations. Due to less negative transfer, we chose homogeneous transfer for our work.

The process of our work includes pretraining, parameter fine-tuning of the original model, and training, which is shown in Figure 4. We applied CAFusionNet proposed in Section 2.4 and trained on dataset A, the source domain. After 150-epoch training and fine-tuning, the pretrained weights were transferred to the triple classification task of dataset B, the target domain.

However, potential negative transfer still exists during the transfer process, which remains a critical issue in this research area. Additionally, subject population and acquisition environment are significantly different between the source domain and target domain in this paper. Parameter fine-tuning is another aspect of transfer learning that requires continual improvement. In this section, the parameter fine-tuning mainly involves adjusting the weights of the top layer of the model. In order to fit with the requirement of triple classification, we set the number of channels in the top dense layer to three, using softmax as the activation function, while the activation function of the final dense layer in the original model is sigmoid. Unlike sigmoid, the output of softmax is a probabilistic vector, which represents the results of forward propagation in multi-classification tasks. Although both dataset A and dataset B share similar feature space, there are differences in factors such as the data collection environment and the types of diseases involved. Specifically, dataset B focuses on the pediatric population, while dataset A includes samples from all age groups. This age distribution difference may lead to significant variations in the physiological and pathological characteristics of heart sounds. In conclusion, while the source and target domain may appear relatively similar in terms of feature space, the potential differences between them should not be overlooked, as there is a certain risk of negative transfer. To enhance the model’s generalization ability for the target domain, we added a residual block before the top layers.

## 3. Results

In order to evaluate that the proposed methods have advantages in terms of accuracy, robustness, and generation ability, our research designed experiments to make a comparison of the advantages of the proposed methods.

### 3.1. Evaluation of CAFusionNet

In this part, we evaluate CAFusionNet using datasets mentioned in Section 2.1. During training, Adam based on adaptive gradient descent algorithm was used for optimizer. Other settings are mentioned in Table 4.

After segmentation and cutting, all signals were split into 7760 sequences which contain 3131 normal signals and 4629 abnormal signals, while each sequence is 5 s in length. Moreover, 6-fold cross validation was used for returning the classification effect of the validated models, which returned six results. The final results of all metrics are the average of the six validation results. In this paper, we made a comparison between the proposed model and the present models. Except for the main structure of the models, no other parameters were changed. Accuracy (Acc), precision (*P*), sensitivity (*Se*), and *F*1-score were used for evaluating the performance of different methods. The theories of precision and sensitivity can be described in (10) and (11).(10)$$P=\frac{TP}{TN+FP}$$(11)$$Se=\frac{TP}{TP+FN}$$
where *TP*, *TN*, *FP*, and *FN* represent the numbers of true positive, true negative, false positive, and false negative in confusion matrices. *F*1-score comprehensively takes into account both of these model capabilities, places greater emphasis on correctly identifying minority classes, and provides a balanced perspective for assessing the overall classification performance on imbalanced datasets.(12)$$F1=\frac{2\cdot P\cdot Se}{P+Se}$$

The comparison results are shown in Table 5.

According to Table 5, the validation results of the traditional models like ResNet have a significant effect on binary classification. However, it shows limitations in accuracy and generalization ability while the feature structure of the heart sound signal is flexible. After fusing feature maps from different convolutional layers, the model performs better. In addition, adding GCT blocks in all feature map outputs from pooling layers also makes a significant increase. In addition, we made a comparison between the present methods and the proposed models with the same datasets and the same basic parameters including learning rate, whose results are shown in Table 6, which shows that the proposed model has a significant effect compared to the present models.

In order to evaluate the most suitable attention block, we made a comparison between the different channel attention modules, adding attention blocks on the same layers without adjusting any other parameters. The final results show that all methods can effectively improve the performance of training tasks, with ECANet performing better than GCT. However, the GCT block contains only three trainable parameters per block, with nine extra parameters in the whole model, while 576 extra parameters exist if we replace all GCT blocks with ECA blocks. Obviously, GCT can significantly reduce the complexity of the training model, especially in the proposed model with numeral channel attention blocks.

### 3.2. Evaluation of Transfer Learning Methods

In this section, we set up three groups of comparative experiments to validate the effectiveness of the transfer learning method for multi-class intelligent diagnosis. Among them, we used CAFusionNet as the standard and trained the model from its initial weights for the traditional machine learning method. Model I and Model II both use the pretrained results of the model proposed in Section 2 on the CinC/Challenge2016 dataset as the starting point for training. The fine-tuning process of the two models is shown in Figure 5.

We adjusted only the parameters of the final fully connected layer based on the original pretrained model to adapt to the new classification task on Model I, while the other parts of the model remain unchanged. Model II evolves further based on Model I by adding a residual module to its structure, which aims to enhance the model’s feature learning capability, especially as dataset A and dataset B have significant differences. The residual module helps the model better capture and learn the critical features required for multi-class classification tasks.

The final results in Table 7 and the output confusion maps (Figure 6) show that the evaluation metrics output by Model I are slightly superior to those of the traditional deep learning methods. This advantage is primarily attributed to the pretrained weights, which have already learned the mapping relationship between the key features of heart sound signals and diagnostic outcomes. When faced with a new training set, the pretrained model leverages the similarity between the source and target domain to quickly adapt to the target domain’s feature space and effectively transfers the weights of each convolutional layer to the training and prediction processes in the target domain. Confusion maps show that the models confused features representing ASD with features representing VSD, resulting in misclassification between both diseases. The transfer learning method reduces this confusion to a certain degree, although this confusion still exists.

Figure 7 shows that Model I converges faster, indicating that during training, Model I can reach a stable state more quickly, thereby improving training efficiency. The classification performance of Model II is slightly better than that of Model I due to the additional convolutional layer, which, to some extent, learns the differences between the target domain and source domain. Combined with the weights accumulated during the pretraining phase, this enhances the model’s generalization ability. However, since some convolutional kernel weights in Model II are trained from their initial state, the model converges more slowly, which increases the model’s complexity and computational cost to a certain extent.

### 3.3. Visualization and Analysis

In this paper, we calculated grad-CAM from 4 points in the proposed models shown in Figure 2 and visualized it by amplitude in the time domain. At first, we visualized the heat map on output from the concatenation layer.

In Figure 8, the *x*-axis stands for the depth of the proposed model, which shows that all reception layers output significant features influencing the predicted outputs of the last dense layer. The following figures describe heat maps from outputs of the third, fourth, and fifth pooling layers, which were weighted by GCT blocks.

In Figure 9, the *x*-axis stands for the time domain space. The first 4s of the gradient values in this figure are visualized, which are fully filled with affective signals. Compared between the three figures, it is clear that the deeper layers contain more abstract features. However, the gradient value outputs from the initial layers show that the gradient envelopes appeared during the whole cardiac circle, which means that the feature maps of the initial layers with high resolution ratios contain more significant features, which are also crucial for diagnosing cardiovascular diseases. However, these features may be ignored with the depth of the model increasing. Additionally, the feature map of the fourth reception layer shows a peak during diastole, which probably means that some key features are contained during diastole.

## 4. Discussion

Our proposed model effectively integrates features from different depths of CNN layers, thereby enriching the feature space represented in the feature maps. The comparison results show that metrics of the testing results increase after concatenating feature maps of the third, fourth, and fifth pooling layers, which confirms that the concatenation of feature maps enhances the robustness and generalization ability of CNN models by leveraging multi-scale information. Further improvements are observed after adding GCT blocks to weight features on the above three layers, which underscores the effectiveness of channel attention mechanisms in weighting significant features relevant to heart valve diseases. Furthermore, the comparison between the proposed model and the present models shows that the proposed models have significant advantages for diagnosing heart valve diseases from complex heart sound features.

We applied homogeneous transfer learning for triple classification, which aimed at solving the problem of insufficient sample sizes in heart sound datasets. Dataset A was used as the source domain to pretrain CAFusionNet. The pretrained weights were then transferred to dataset B. The test results indicate that the pretrained and fine-tuned model exhibited strong generalization ability.

We proposed a heat map based on grad-CAM to visualize the process of the proposed model. The visualization results indicate significant differences in the gradient values and their distribution in the time domain across different depths of ResNet. The shallow feature maps exhibit a more dispersed envelope, possibly indicating that the model captures broader and more fundamental signal features in the early stages of the network. In contrast, the deep feature maps are concentrated in the S1 and S2 phases, suggesting that the model learns more refined and representative cardiac activity features in the deeper layers. The Grad-CAM-based visualization heatmaps confirm, from a gradient perspective, the influence of network depth on the feature space. They also demonstrate the effectiveness of multi-scale feature fusion in expanding the feature space and enhancing classification performance.

## 5. Conclusions

This paper proposes CAFusionNet, which applies channel attention models before fusing features from different layers, for heart sound classification. The model paid attention to features representing heart valve diseases by focusing weights on more significant channels, and the evaluating results show that the proposed model performed better than the present models, especially while adding GCT blocks on all layers. We transferred weights from CAFusionNet pretrained by public heart sound datasets to classify heart sound signals labeled with different types of heart valve diseases and “normal”. The final results are improved, which can be contributed to the accumulating weights in pretraining.

We visualized output features by heat maps based on Grad-CAM, which indicate that significant features are significantly different between the feature maps at different resolution ratios. Thus, fusing features from different layers is necessary.

However, our research has some limitations. The end-to-end method we proposed, whose performance is still highly dependent on model parameters and the quality of datasets, has room for enhancing robustness. We will explore manual feature extraction methods to solve this problem of end-to-end methods in the future. Furthermore, the quality of the source domain used for homogeneous transfer learning has influence on the results; therefore, we will investigate cross-domain transfer learning methods pretrained on large-scale datasets.

## Figures and Tables

**Figure 1 bioengineering-12-00290-f001:**
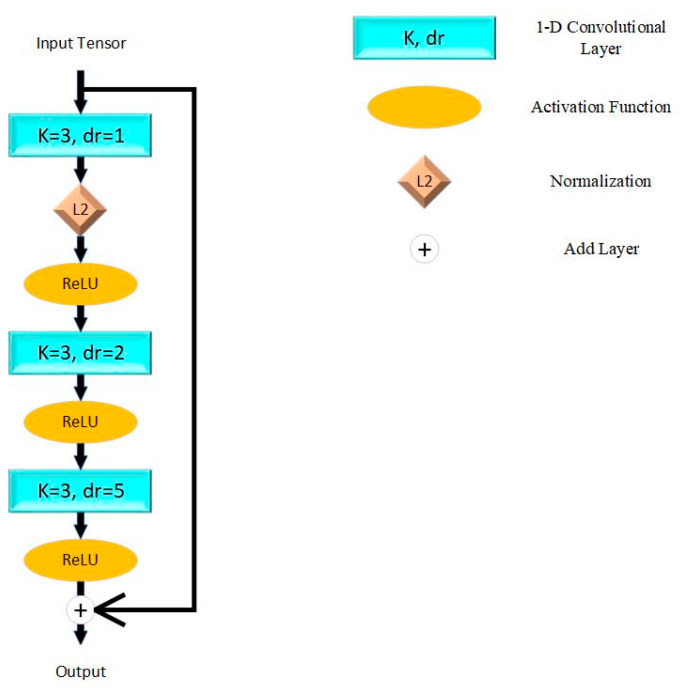
Residual block in proposed model. K = kernel size; dr = dilation rate; L2 = L2 normalization; ReLU = Rectified Linear Unit.

**Figure 2 bioengineering-12-00290-f002:**
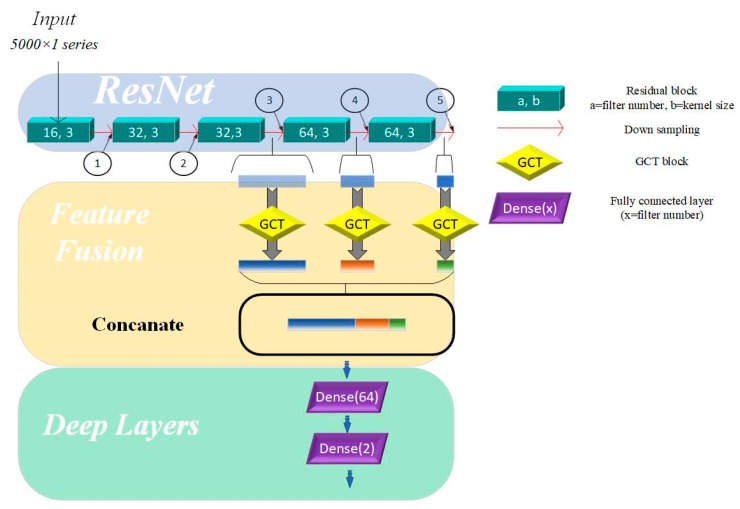
Multi-scale model based on channel attention mechanism. Directions of arrows are directions of forward propagation; GCT = Gated Channel Transformation.

**Figure 3 bioengineering-12-00290-f003:**
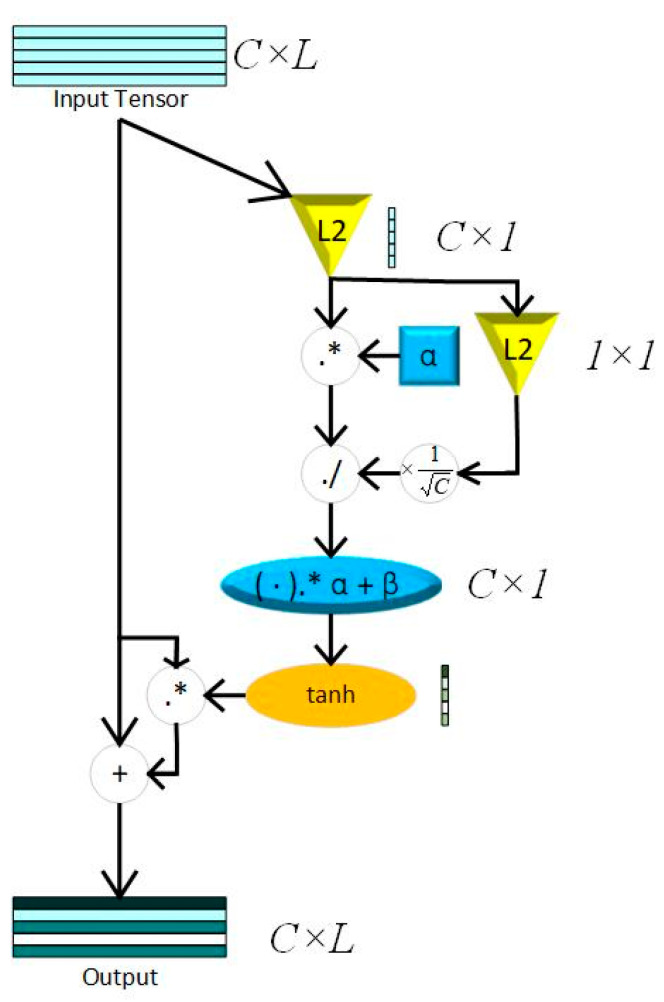
GCT block; L2 = L2 normalization; *C* = channel; *L* = Feature Length; “.*” represents element-wise multiplication; “./” represents element-wise division.

**Figure 4 bioengineering-12-00290-f004:**
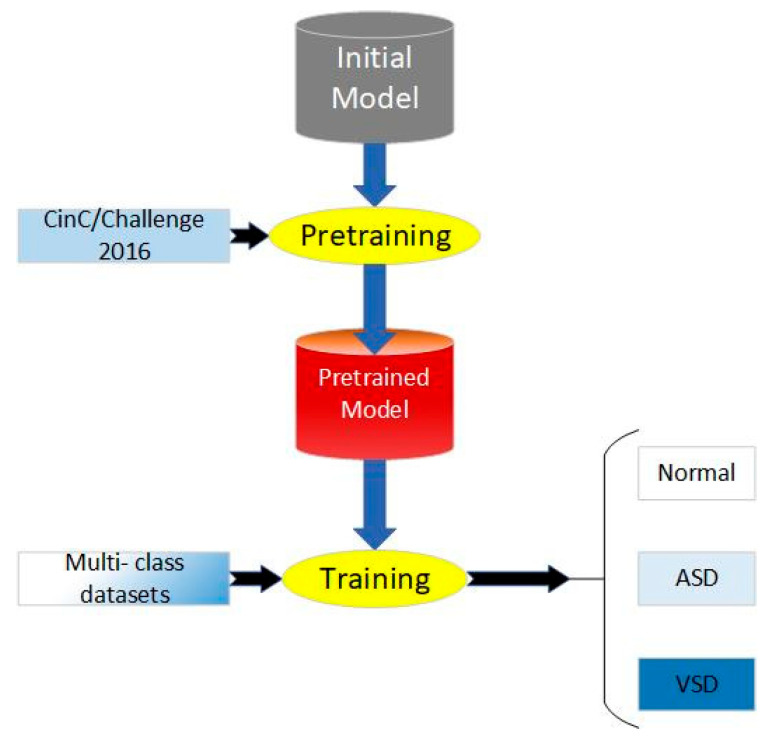
Process of proposed transfer learning method.

**Figure 5 bioengineering-12-00290-f005:**
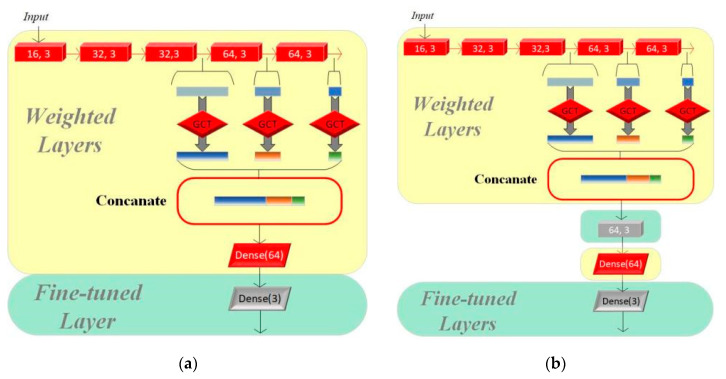
(**a**) Structure of Model I. **(b**) Structure of Model II.

**Figure 6 bioengineering-12-00290-f006:**
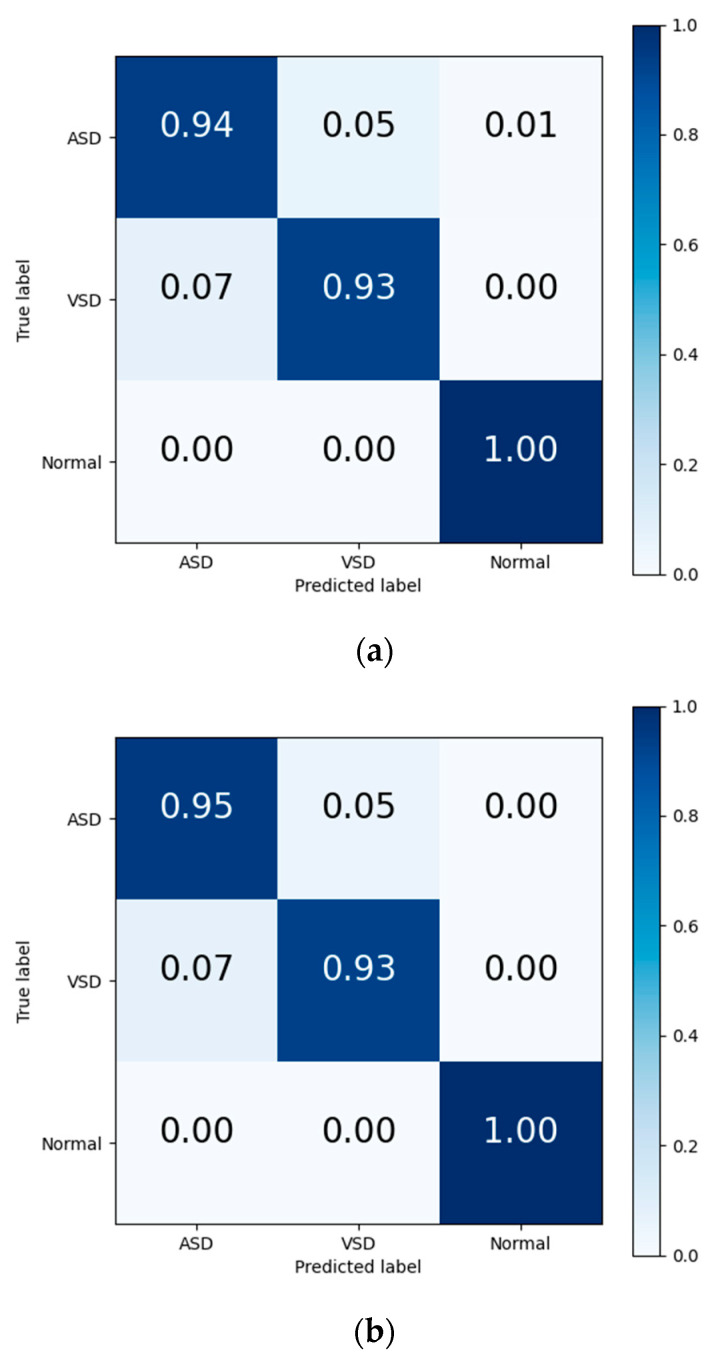
Confusion matrix of evaluation results from (**a**) traditional DL, (**b**) Model I, and (**c**) Model II.

**Figure 7 bioengineering-12-00290-f007:**
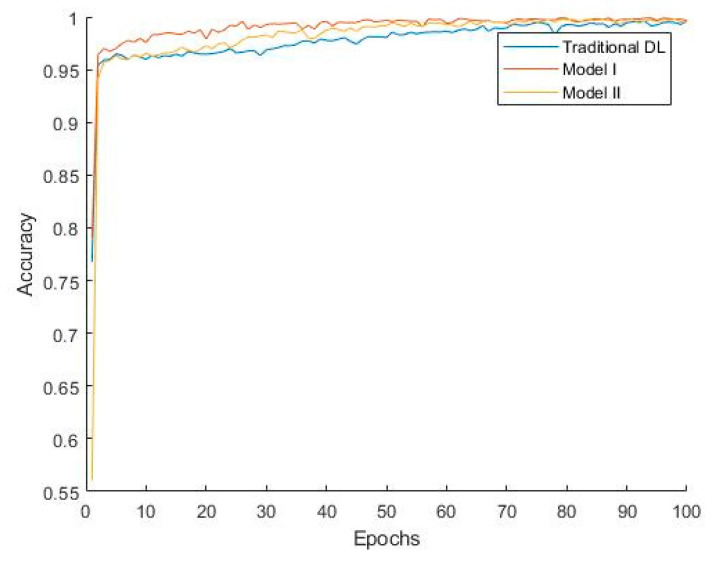
Convergence curves of different methods.

**Figure 8 bioengineering-12-00290-f008:**
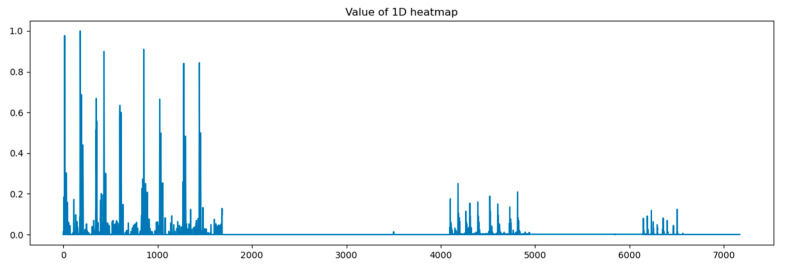
Heat map of concatenation layer in proposed model.

**Figure 9 bioengineering-12-00290-f009:**
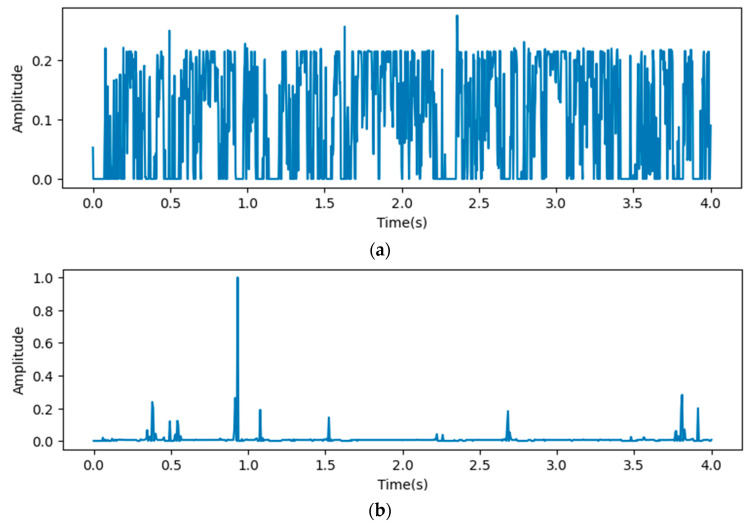
Heat map outputs from point 3 (**a**), 4 (**b**), and 5 (**c**) of proposed model.

**Table 1 bioengineering-12-00290-t001:** Components of heart sound signal.

Component	Origin	Period	Notes
S1	closing of mitral valve and tricuspid valve at the beginning of systole	Beginning of systole	Fundamental heart sound
S2	closing of aortic valve and pulmonary valve at the beginning of diastole	Beginning of diastole	
S3	rapid filling of ventricle	Diastole	Cannot be detected from healthy adult
S4	atria contract push blood into stiff or non-compliant ventricle	Diastole	

**Table 2 bioengineering-12-00290-t002:** Datasets applied in this study.

Dataset	Trial Population	Labels	Total Number	Total Time/h
Dataset A	All ages	Normal/Abnormal	3240	20.216
Dataset B	4M–16Y	Normal/ASD/VSD	268	1.414

**Table 3 bioengineering-12-00290-t003:** Conditions of different depths in residual networks.

Point (Shown in Figure 2)	1	2	3	4	5
Receptive Field Size	6	16	40	100	204
Length of Feature Map per Circle	500	250	125	62	31

**Table 4 bioengineering-12-00290-t004:** Settings of hyper-parameters in experiments.

Parameters	Setting
Optimizer	Adam
Learning Rate	0.001
Loss Function	Binary Cross Entropy (BCE)
Training Epochs	150
Batch Size	32

**Table 5 bioengineering-12-00290-t005:** Comparison of model performance on binary classification task.

Models	Acc	P	Se	F1-Score
ResNet ^1^	0.8839	0.8836	0.8844	0.8840
ResNet+Fusion ^2^	0.8951	0.8972	0.8958	0.8965
Proposed model	0.9323	0.9323	0.9323	0.9323
MsTGANet [27]	0.8906	0.8873	0.9936	0.8905
LeLWTNet [28]	0.8482	0.8482	0.8482	0.8482
WaveNet [29]	0.8467	0.8467	0.8467	0.8467
Attention U-Net [30]	0.8594	0.8595	0.8494	0.8594

^1^ Proposed model without feature fusion structure or GCT blocks. ^2^ Proposed model without GCT blocks.

**Table 6 bioengineering-12-00290-t006:** Comparison between different channel attention modules.

Attention Module	Acc	P	Se	F1-Score
SENet	0.9204	0.9198	0.9204	0.9200
ECANet	0.9330	0.9338	0.9330	0.9334
GCT (Proposed model)	0.9323	0.9323	0.9323	0.9323

**Table 7 bioengineering-12-00290-t007:** Comparison of testing metrics between transfer learning and traditional ML methods.

Methods	Acc	P	Se	F1-Score
Traditional DL	0.9696	0.9609	0.9606	0.9607
Transfer Learning (Model I)	0.9652	0.9655	0.9650	0.9652
Transfer Learning (Model II)	0.9665	0.9667	0.9665	0.9666

## Data Availability

Data supporting the reported results can be obtained on request.

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
