# Peer review of "Heart Sound Classification Based on Multi-Scale Feature Fusion and Channel Attention Module"

_bioengineering, 2025, doi:10.3390/bioengineering12030290_

Round 1
Reviewer 1 Report
Comments and Suggestions for Authors
Reviewer’s comment
Review of submission to bioengineering
Title: Heart Sound Classification Based on Multi-scale Feature Fusion and Channel Attention
Module
Authors: M. Li et al.
The authors propose a heart sound classification model to fuse features from different layers. Its performance is better than that of some existing models in the selected databases.
Generally, the reviewer recommends that this paper should be accepted with minor modifications because of its good contributions. The following comments may help the authors to improve the quality of the paper:
âš« There are many careless and grammar mistakes in the paper. The authors should employ a
native English speaker or make use of some AI tools to proofread the whole paper carefully.
Here are some:
â—¼ On page 5, line 179, “every layers” should be “every layer”.
â—¼ On page 5, line 183, “5 residual blocks” should be “five residual blocks”.
â—¼ On page 12, line 362, the reviewer wonders if the title should be “Choose the sample in
the validation set with the lowest BCE value.”
âš« A diagram should describe S1, S2, S3, and S4 in a cycle of a heart sound signal.
âš« The symbols used in Figures 1, 2, 3, and 4 should be described.
â—¼ What does K, dr, and ReLU mean in Figure 1?
â—¼ What do the numbers in the inception block mean in Figure 2?
â—¼ What does GCT and Dense(?) mean in Figure 2?
â—¼ Are the L2 in Figures 1 and 3 are the same?
â—¼ What is the meaning in Figure 4 when the blue arrow hits the black arrow?
âš« A block diagram should give the full picture of the proposed system. Figures 1 to 4 should be four blocks, and they should be found in this block diagram.
âš« The reviewer wonders if 3.3 should be in Section 2 because it seems related to the
implementation.
âš« Is the discussion a part of the conclusion?
âš« The reviewer does not understand the problem described in line 446, page 15. Can the authors tell more?
Check the attached document
Author Response
Comment 1: “There are many careless and grammar mistakes in the paper. ”.
Response 1: Thank you for pointing this out. We agree this comment. Therefore, we checked the manuscript and corrected all error about grammar and tense.
Comment 2: “ A diagram should describe S1, S2, S3, and S4 in a cycle of a heart sound signal.”
Response 2: Thank you for pointing this out. We agree this comment. Therefore, we made a diagram (Table 1) to descride the 4 components of a cardiac circle. The contents of Table are based on line 38-51, page 2-3 of original manuscript.
Comment 3: “The symbols used in Figures 1, 2, 3, and 4 should be described.”
Response 3: Thank you for pointing this out. We agree this comment. Therefore, we clarified the meanings of K, dr, L2 and ReLU in the caption of Figure1. We clarified the meanings of number in residual blocks and “Dense()” in the Figure 2 in the legend. We clarified the representation of GCT in the caption of Figure 2, meanwhile the full name of GCT has been represented in line 135, page 2.
Comment 4: “ A block diagram should give the full picture of the proposed system. Figures 1 to 4 should be four blocks, and they should be found in this block diagram.”
Response 4: Thank you for pointing this out. Figure 2 is the full picture of proposed model. Figure1 and Figure 3 are units of proposed model. Figure 4 is the process of proposed transfer learning method.
Comment 5: “ The reviewer wonders if 3.3 should be in Section 2 because it seems related to the implementation.”
Response 5: Thank you for pointing this out. We agree this comment. Therefore, we placed 3.3 to 2.4.
Comment 6: “Is the discussion a part of the conclusion?”
Respense 6: Thank you for pointing this out. In our manuscript, the Discussion section is intended to provide a detailed interpretation of our experimental results, compare our findings with existing literature. The Conclusion section offers a concise summary of the main contributions of our study, outlines limitations of our work as well as directions for future research.
Comment 7: “The reviewer does not understand the problem described in line 446, page 15. Can the authors tell more?”
Response 7: Thank you for pointing this out. We agree this comment. The limitations and future works in conclusion have been majorly revised following the report of another reviewer. The problem mentioned in conclusion means that the performance of proposed end-to-end method is highly depends on model parameters and the quality of datasets.
Reviewer 2 Report
Comments and Suggestions for Authors
Summary of the article under consideration: The paper presents CAFusionNet, a model for classifying heart sounds. It combines features from different layers and uses channel attention to improve accuracy. The model also uses transfer learning to handle small datasets.
General comments: The article addresses an interesting problem and presents experimental results that are noteworthy. However, it requires several improvements in both presentation and scientific significance. Here are some suggestions for further improvement:
(1) The abstract is fine, but the introduction section requires to be revised comprehensively. To improve the introduction section of the article, consider the following recommendations to present the information in a coherent and structured manner:
a. Begin with a brief overview of heart sound classification, explaining its importance in diagnosing cardiovascular diseases (CVDs).
b. Mention the prevalence and impact of CVDs globally, highlighting the need for accurate and efficient diagnostic methods. In other words, identify the particular research problem in heart sound classification and provide motivation for it. Clearly state the specific problem within heart sound classification that the research aims to address. Provide motivation by discussing the limitations of traditional diagnostic methods, such as reliance on clinical expertise and the associated precision rates.
c. How the target research problem is being addressed by state of the art. Summarize current state-of-the-art approaches in heart sound classification, focusing on deep learning and convolutional neural networks (CNNs). Highlight key advancements and their contributions to the field.
d. Discuss the limitations of existing methods, such as issues with dataset size, model generalization, and the need for improved accuracy and robustness. Mention any specific challenges related to noise, variability in recording conditions, and computational complexity.
e. Introduce the proposed solution, CAFusionNet, and outline its major components, such as multi-scale feature fusion and channel attention mechanisms. Briefly explain how these components are designed to address the identified limitations. Provide a brief explanation of how multi-scale feature fusion and channel attention mechanisms enhance the model's performance. Discuss the role of transfer learning in mitigating issues related to small datasets and improving model generalization.
f. Describe the validation process, including the selection of datasets (PhysioNet/CinC Challenge 2016 and a proprietary pediatric dataset). Justify the choice of these datasets and explain how they contribute to the robustness and reliability of the study. Summarize the key results, such as the achieved accuracy of 0.9323, and compare it with existing models.
(2) To enhance the clarity and comprehensiveness of the article, it is recommended to include a dedicated background section. This section should provide all the necessary background information to help readers understand the context and significance of the research. Specifically, the background section should cover the following points:
a. Overview of Heart Sound Classification: Explain what heart sound classification is and its importance in diagnosing cardiovascular diseases (CVDs). Discuss the physiological basis of heart sounds, including the origins of S1, S2, S3, and S4 sounds, and their relevance to heart valve diseases.
b. Current Diagnostic Methods: Describe traditional methods of heart sound diagnosis, highlighting their reliance on clinical expertise and the associated precision rates. Mention the limitations of these traditional methods, such as subjectivity and the need for more reliable, automated tools
c. Advancements in Deep Learning for Heart Sound Classification: Provide an overview of how deep learning, particularly Convolutional Neural Networks (CNNs), has been applied to heart sound classification. Summarize key advancements in the field, including multi-scale feature fusion and channel attention mechanisms.
By including a dedicated background section with these points, the article will provide readers with a comprehensive understanding of the research context, the significance of the problem being addressed, and the innovative aspects of the proposed solution. This will enhance the overall readability and impact of the article.
(3) It is recommended to include a dedicated Related Work section. This section should compare the proposed method with state-of-the-art approaches in terms of various attributes. A comparison table should be included to clearly present the differences and similarities between the methods. This will help in judging the novelty and effectiveness of the proposed solution. In other words, this section should provide the most relevant and recent methods in heart sound classification. Highlight the key features, strengths, and limitations of each method. Compare the proposed CAFusionNet with state-of-the-art methods in terms of various attributes such as accuracy, robustness, computational efficiency, dataset size, and generalization ability. Discuss how each method addresses the challenges in heart sound classification. Create a table that lists the state-of-the-art methods and compares them based on the identified attributes. Include columns for method name, key features, accuracy, dataset size, computational requirements, and any other relevant attributes. Summarize the key points from the related work section. Reinforce the novelty and significance of the proposed method.
(4) Provide a high-level overview of the CAFusionNet architecture, including its main components and their roles. Use a block diagram to illustrate the overall structure of the model. Describe the input layer, including the type and dimensions of the input data (e.g., heart sound signals). Explain the role of convolutional layers in feature extraction. Mention the number of layers, kernel sizes, and activation functions used. Discuss the purpose of pooling layers for downsampling and reducing computational complexity. Describe how features from different layers with varying resolution ratios are fused. Use a diagram to show the interconnections between layers. Explain the Gated Channel Transformation (GCT) blocks and their role in weighting key features. Include a detailed diagram of a GCT block. Describe the fully connected layers and their role in classification. Mention the activation functions used (e.g., ReLU, softmax). To summarize, break down the model into functional components and describe the role of each component. Use a hierarchical diagram to show different levels of the system structure, from high-level components to detailed sub-components.
(5) Describe the flow of data through the model, from input to output. Use flowcharts or sequence diagrams to illustrate the data flow and the interactions between different components. Discuss the training process, including the loss function (e.g., Binary Cross Entropy), optimizer (e.g., Adam), learning rate, and number of epochs. Explain how the model parameters are updated during training and the role of backpropagation.
(6) Describe the validation process, including the use of cross-validation and performance metrics (e.g., accuracy, precision, sensitivity, F1-score).
(7) Explain how the model's performance is evaluated and compared with state-of-the-art methods.
(8) In experimental part, experimental settings must be explicitly described. A block diagram of experimental settings must be provided. The results must be visualized. The results must be compared with recent state of the art methods in terms of various performance attributes. These attributes must be clearly explained before discussing the performance comparison. The limitations of the proposed method must be described.
(9) More recent works should be cited and the results must be compared with more recent works (past 3 to 4 years)
(10) The conclusion section should focus on what has been presented and what has been achieved. It is better to connect existing achievement with some future directions.
Author Response
Comment 1: “The abstract is fine, but the introduction section requires to be revised comprehensively. To improve the introduction section of the article, consider the following recommendations to present the information in a coherent and structured manner:”
Response 1: Thank you for pointing this out. We agree this comment. Therefore, we changed the structure of introduction about background, related works and proposed works.
Comment 2: “To enhance the clarity and comprehensiveness of the article, it is recommended to include a dedicated background section. This section should provide all the necessary background information to help readers understand the context and significance of the research. Specifically, the background section should cover the following points:”
Response 2: Thank you for pointing this out. We agree this comment. We set a part (1.1) under section “Introduction” to describe background information about the overview of CVDs, cardiac circle and the significance heart sound classification. Meanwhile, we use a diagram (Table1.1) To describe the four components of heart sound following the report of another reviewer.
Comment 3: “ It is recommended to include a dedicated Related Work section. This section should compare the proposed method with state-of-the-art approaches in terms of various attributes. A comparison table should be included to clearly present the differences and similarities between the methods. This will help in judging the novelty and effectiveness of the proposed solution. In other words, this section should provide the most relevant and recent methods in heart sound classification. Highlight the key features, strengths, and limitations of each method. Compare the proposed CAFusionNet with state-of-the-art methods in terms of various attributes such as accuracy, robustness, computational efficiency, dataset size, and generalization ability. Discuss how each method addresses the challenges in heart sound classification. Create a table that lists the state-of-the-art methods and compares them based on the identified attributes. Include columns for method name, key features, accuracy, dataset size, computational requirements, and any other relevant attributes. Summarize the key points from the related work section. Reinforce the novelty and significance of the proposed method.”
Response 3: Thank you for pointing this out. We set a part (1.2) under section “Introduction” to provide the most relevant and recent methods in heart sound classification. The comparison data between proposed method and present works is shown in Table 5.
The limitations, datasets and methods of recent works are logically summarized in text. However, computational requirement, sample numbers and some relevant attributes are not mentioned and hard to combined in a table. Additionally, the text highlights contents related to proposed methods. Thus, we think text can effectively summarize related works rather than a table.
Comment 4: “Provide a high-level overview of the CAFusionNet architecture, including its main components and their roles. Use a block diagram to illustrate the overall structure of the model. Describe the input layer, including the type and dimensions of the input data (e.g., heart sound signals). Explain the role of convolutional layers in feature extraction. Mention the number of layers, kernel sizes, and activation functions used. Discuss the purpose of pooling layers for downsampling and reducing computational complexity. Describe how features from different layers with varying resolution ratios are fused. Use a diagram to show the interconnections between layers. Explain the Gated Channel Transformation (GCT) blocks and their role in weighting key features. Include a detailed diagram of a GCT block. Describe the fully connected layers and their role in classification. Mention the activation functions used (e.g., ReLU, softmax). To summarize, break down the model into functional components and describe the role of each component. Use a hierarchical diagram to show different levels of the system structure, from high-level components to detailed sub-components.”
Response 4: Thank you for pointing this out. We agree this comment. The overview of proposed method has been provided in 2.3, whose text and diagrams are revise:
The type and dimensions of the input layer, parameters in the models, interconnection of each layers are mentioned in diagrams (Figure 1-2).
The roles of convolutional layers and pooling layers are mentioned.
The fusion method, activation functions are mentioned.
Comment 5: “Describe the flow of data through the model, from input to output. Use flowcharts or sequence diagrams to illustrate the data flow and the interactions between different components. Discuss the training process, including the loss function (e.g., Binary Cross Entropy), optimizer (e.g., Adam), learning rate, and number of epochs. Explain how the model parameters are updated during training and the role of backpropagation.”
Response 5: Thank you for pointing this out. We agree this comment. Therefore, flow of data is mentioned in Figure 1-3. Training process is mentioned in line 348-358, page 10.
Comment 6: “Describe the validation process, including the use of cross-validation and performance metrics (e.g., accuracy, precision, sensitivity, F1-score).”
Response 6: Thank you for pointing this out. We agree this comment. Therefore, we add the theories of each performance metrics in line 359-370, page 10-11.
Comment 7: “ Explain how the model's performance is evaluated and compared with state-of-the-art methods.”
Response 7: Thank you for pointing this out. We agree this comment. The data of performance is mention in Table 5. In Table 5, the first 3 rows are control experiment of proposed model, the last 4 rows are comparison between proposed model and present models.
Comment 8: “ In experimental part, experimental settings must be explicitly described. A block diagram of experimental settings must be provided. The results must be visualized. The results must be compared with recent state of the art methods in terms of various performance attributes. These attributes must be clearly explained before discussing the performance comparison. The limitations of the proposed method must be described. ”
Response 8: Thank you for pointing this out. We agree this comment. Therefore we created a table (Table 4) to mention experiment setting. Results are visualized with convergence curves, heat maps and confusion matrices in Figure 6-9. The limitation of proposed works are discribed in Conclusion.
Comment 9: “ More recent works should be cited and the results must be compared with more recent works (past 3 to 4 years)”
Response 9: Thank you for pointing this out. We agree this comment. Therefore, we add extra two recent works to make a comparison shown in Table 5.
Comment 10: “The conclusion section should focus on what has been presented and what has been achieved. It is better to connect existing achievement with some future directions.”
Response 10: Thank you for pointing this out. We agree this comment. Therefore, we revised the limitations and future works in Conclusion.
Reviewer 3 Report
Comments and Suggestions for Authors
This study tries to classify heart sound based on multi-scale feature fusion and channel attention module. After careful reading of this manuscript, I have the following comments:
(1) The study has its significant contribution to the field.
(2) The proposed method is workable and the model is good established.
(3) The manuscript is good written and discussed.
(4) I will suggest the authors to compare their method with the other researchers to discuss the pros and cons.
(5) Please give some clinical application of this research.
Author Response
Comment 1: “ I will suggest the authors to compare their method with the other researchers to discuss the pros and cons.”
Respense 1: Thank you for pointing this out. We agree this comment. The comparison of proposed work and present works is describe in Table 5. In Table 5, the first 3 rows are control experiment of proposed model, the last 4 rows are comparison between proposed model and present models.
Comment 2: “Please give some clinical application of this research.”
Respense 2: Thank you for pointing this out. We agree this comment. Therefore, we add the clinical significance of proposed research in line 147-151, page 4.
Round 2
Reviewer 2 Report
Comments and Suggestions for Authors
The raised comments have been addressed.